# Bioinformatics Prediction and Experimental Verification Identify CAB39L as a Diagnostic and Prognostic Biomarker of Kidney Renal Clear Cell Carcinoma

**DOI:** 10.3390/medicina59040716

**Published:** 2023-04-06

**Authors:** Yunfei Wu, Zhijie Xu, Xiaoyi Chen, Guanghou Fu, Junjie Tian, Yue Shi, Junjie Sun, Baiye Jin

**Affiliations:** 1Department of Urology, The First Affiliated Hospital, School of Medicine, Zhejiang University, Hangzhou 310009, China; 2Zhejiang Engineering Research Center for Urinary Bladder Carcinoma Innovation Diagnosis and Treatment, Hangzhou 310024, China

**Keywords:** kidney renal clear cell carcinoma, prognosis, diagnosis

## Abstract

*Background and Objectives*: Calcium-binding protein 39-like (CAB39L) has been reported to be downregulated and possessed diagnostic and prognostic values in several types of cancer. However, the clinical value and mechanism of CAB39L in kidney renal clear cell carcinoma (KIRC) remain unclear. *Materials and Methods*: Bioinformatics analysis was conducted using different databases including TCGA, UALCAN, GEPIA, LinkedOmics, STRING, and TIMER. One-way variance analysis and *t*-test were chosen to investigate the statistical differences of CAB39L expression in KIRC tissues with different clinical characteristics. The receiver operating characteristic (ROC) curve was chosen to assess the discriminatory capacity of CAB39L. Kaplan–Meier curves were employed for assessing the influence of CAB39L on the progression-free survival (PFS), disease-specific survival (DSS), and overall survival (OS) of KIRC patients. The independent prognostic significance of clinical parameters for OS such as CAB39L expression in KIRC patients was estimated by Cox analysis. A series of in vitro functional experiments and Western blot (WB) and immunohistochemistry (IHC) were used to validate the relative protein expression and function of CAB39L. *Results*: The mRNA and protein levels of CAB39L were relatively downregulated in KIRC samples. Meanwhile, hypermethylation of the CAB39L promoter region was possibly associated with its low expression in KIRC. The ROC curve showed that the mRNA expression of CAB39L had a strong diagnostic value for both early and late KIRC. Kaplan–Meier survival curves indicated that a higher mRNA level of CAB39L predicted good PFS, DSS, and OS. The mRNA expression of CAB39L was an independent prognostic factor (hazard ratio = 0.6, *p* = 0.034) identified by multivariate Cox regression analysis. The Kyoto Encyclopedia of Genes and Genomes (KEGG) and Gene Ontology (GO) analysis exhibited that CAB39L was mainly associated with substance and energy metabolism. Finally, overexpression of CAB39L impaired the proliferation and metastasis of KIRC cells in vitro. *Conclusions*: CAB39L possesses prognostic and diagnostic capacity in KIRC.

## 1. Introduction

Renal cell carcinoma (RCC) is known as one of the most common malignant cancers worldwide, ranking as the sixth most common cancer in males and the ninth in females [1]. Patients with kidney renal clear cell carcinoma (KIRC) account for nearly 80% of RCC patients; therefore, it is regarded as the most common histological subtype [1,2]. As KIRC symptoms in the early stage are insidious, approximately 30% of patients with KIRC are diagnosed in the late stage [3]. Moreover, most patients with KIRC are resistant to chemotherapy and radiotherapy, and surgery becomes the optimal treatment for KIRC [2,4,5]. Although the treatment of KIRC has been enriched by immunotherapy and targeted therapy, patients with advanced and metastatic KIRC who have lost the opportunity to have surgery still have dismal outcomes [6,7,8]. Hence, there is an urgent demand for discovering new sensitive biomarkers and therapeutic targets to predict and enhance the prognosis of KIRC.

Calcium-binding protein 39-like (CAB39L) is a β-isoform of CAB39 and a scaffold protein that combines and stabilizes the liver kinase B1 (LKB1) protein in an activated conformation necessary for the phosphorylation of substrates [9,10]. More and more evidence is emerging to indicate that CAB39L plays an anti-tumor role in many cancers, such as breast cancer and gastric cancer [11,12], but the clinical significance and function of CAB39L in KIRC have not been reported.

In this study, bioinformatics analysis was executed to identify the effect of CAB39L in KIRC tumorigenesis and prognosis. The results found that KIRC tissues had significantly lower CAB39L expression than normal kidney tissues, which was associated with poor prognosis and advanced clinical stage. It was also noticed that CAB39L had a strong diagnostic value regardless of early or late KIRC. In addition, the overexpression of CAB39L in KIRC had an impact on its proliferation, migration, and invasion. To sum up, CAB39L may be applied as a promising diagnostic and prognostic biomarker in KIRC.

## 2. Materials and Methods

### 2.1. Data Resource and Processing

Level 3 gene expression profiles (workflow type: HTSeq-FPKM) were downloaded from the KIRC dataset on The Cancer Genome Atlas (TCGA) portal (https://portal.gdc.cancer.gov/, accessed on 3 March 2022), including 539 KIRC specimens and 72 corresponding tumor-adjacent tissues. HTSeq-FPKM values were transformed into TPM values for further analysis. Meanwhile, the corresponding clinicopathological data of KIRC patients were extracted from TCGA. For the pan-cancer analysis, UCSC Xena (https://xenabrowser.net/, accessed on 3 March 2022) was used to download RNA-Seq data containing 33 cancer types from Genotype-Tissue Expression (GTEx) and TCGA database. The median mRNA expression of CAB39L was defined as a cut-off to sort KIRC patients.

### 2.2. Comprehensive Analysis

TIMER (https://cistrome.shinyapps.io/timer/, accessed on 3 March 2021) [13] was chosen to explore the mRNA expression of CAB39L in KIRC samples and normal adjacent samples.

GEPIA (http://gepia.cancer-pku.cn/, accessed on 3 March 2021) [14] is a comprehensive website that analyzes and visualizes the RNA data of different tumors and normal samples. On this basis, the mRNA expression of CAB39L was compared between KIRC and normal samples in this study. The mRNA expression of CAB39L at different pathologic stages was also evaluated.

UALCAN (http://ualcan.path.uab.edu/index.html/, accessed on 3 March 2021) [15] is a bioinformatics platform that provides easy access to investigate the relationship between gene expression (mRNA or protein) and clinicopathological data. In this study, the promoter methylation of CAB39L was also evaluated based on different clinicopathological features.

LinkedOmics (http://www.linkedomics.org/admin.php/, accessed on 4 March 2021) [16] is an online platform that processes multi-omics data from TCGA database. In this study, CAB39L co-expressed genes in KIRC were searched and examined by GO and KEGG pathway enrichment analysis.

HPA (https://www.proteinatlas.org/, accessed on 4 March 2021) [17] was applied to compare the CAB39L expression between KIRC specimens and normal kidney tissue samples.

MethSurv (https://biit.cs.ut.ee/methsurv/, accessed on 4 March 2021) [18] was applied to explore the methylation status of different CpG sites located at the CAB39L gene promoter region.

### 2.3. Screening of Differentially Expressed Genes (DEGs) and Functional Enrichment Analysis 

The R package “DESeq2” [19] was run to filter DEGs between KIRC samples with high and low CAB39L expression. |logFC| ≥ 2 and *p* < 0.05 were defined as the filter criteria. The results of DEGs were displayed in a volcano plot using the R package “ggplot2”. The top 15 downregulated and upregulated CAB39L-related DEGs were further chosen to perform GO and KEGG analysis by running the “clusterProfiler” package [20].

### 2.4. Interaction Analysis

STRING (https://string-db.org/, accessed on 4 March 2021) [21] was applied to establish a protein–protein interaction (PPI) network of CAB39L. The prediction pathways of CAB39L and its associated proteins were also analyzed by GO and KEGG.

### 2.5. Cell Culture and Transfection

Human KIRC cells A-498 and 786-O were purchased from ATCC. All cells were cultured in MEM medium (Gibico, Waltham, MA, USA) containing 10% FBS (Sturgeon Bay, WI, USA). The CAB39L overexpression vector and the empty control vector were bought from RiboBio (Guangzhou, China), and corresponding vectors mixed with jetPRIME transfection reagent (Polyplus, Strasbourg, France) were applied to transfect cells according to the manufacturer’s instructions. Here, 30–40% confluence off cells in 6 cm dishes was cultured in 3 mL complete medium and transfection mixtures (2 μg pcDNA3.1 empty vectors/pcDNA3.1 CAB39L-overexpression vectors, transfection reagent A 300 μL, and transfection reagent B 4 μL) for 48 h before in vitro functional experiments.

### 2.6. In Vitro Functional Experiments

The cell proliferation capacity was evaluated using colony formation assay and cell counting kit 8 (CCK-8). Here, 2000 pretreated cells were seeded in each well of the 96-well plate for the CCK-8 assay. The absorbance value at 450 nm was determined after the addition of 10 μL CCK-8 reagent (MCE, NJ, USA) in each well for 1 h at 37 °C. As for the colony formation assay, 500 pretreated KIRC cells were seeded in each well of the six-well plate for 2 weeks. Relative colony rates were calculated after staining with 0.2% crystal violet and fixing in 4% paraformaldehyde. The cell migration and invasion abilities were measured using the transwell assay. Then, 200 μL serum-free DMEM medium with 2 × 104 pretreated cells were injected into the upper transwell chamber (Corning, New York, NY, USA) for migration assay. Similarly, cells were seeded in the upper transwell chamber precoated with Matrigel (Corning, New York, NY, USA) for invasion assay. As an attractant, 700 μL DMEM medium (10% FBS) was injected into the lower chamber. Migration or invasion assays were also subsequently treated with 4% paraformaldehyde and 0.2% crystal violet after 48 h.

### 2.7. Western Blot (WB)

Cells were mixed with RIPA lysis buffer (Fdbio, Hangzhou, China) and underwent 6 s ultrasonic disruption for complete lysing. The concentration of protein following 12,000× *g* centrifugation was quantified by a BCA protein assay kit (Fdbio, Hangzhou, China). Proteins were subsequently divided by 10% SDS-polyacrylamide gel electrophoresis and shifted to the PVDF membrane (Millipore, Burlington, MA, USA). The cropped PVDF membrane according to protein marker (Vazyme, Nanjing, China) was blocked with 5% fat-free milk for 1 h and then cultured with primary antibodies at 4 °C overnight. Primary antibodies were prepared in 5% BSA solution. BSA was dissolved in TBST. After washing with premade TBST buffer (Fdbio, Hangzhou, China) three times per 15 min, the membranes were cultured with secondary antibody conjugated with HRP (1:4000, Fdbio, Hangzhou, China) at room temperature for 1 h. Finally, after washing with TBST buffer three times again, protein expression on the PVDF membranes was measured using the FDbio-Dura ECL Kit on the Bio-Rad CD touch detection system. Anti-α-Tubulin (1:1000, Beyotime, Shanghai, China) and anti-CAB39L (1:1000, Abclonal, Wuhan, China) were the antibodies used in this study.

### 2.8. KIRC Tissue Samples and Immunohistochemistry (IHC)

KIRC and paired paracancerous samples were gathered from patients who received radical total nephrectomy at the First Affiliated Hospital, Zhejiang University School of Medicine. The Ethics Committee of the First Affiliated Hospital, Zhejiang University School of Medicine authorized this procedure. Patients enrolled in this experiment all signed written informed consent. Immunohistochemical staining was applied to measure the CAB39L protein level in KIRC and paired paracancerous samples using an anti-CAB39L antibody (1:200, Proteintech, Wuhan, China). The details of this experiment were described in our previous study in detail [22].

### 2.9. Statistical Analysis

R software (version 4.1.1, Microsoft, Washington, DC, USA) and GraphPad Prism 9.0 software (GraphPad Software, Boston, MA, USA) were both used for statistical analysis. Statistical differences were analyzed using one-way variance analysis and student’s *t*-test. The association between CAB39L expression and clinicopathological data was analyzed using Pearson’s chi-square test. The ROC curve by the pROC package [23] was applied to assess the potential diagnostic value of CAB39L expression and acquire the area under the curve (AUC). AUC > 0.7 displayed good diagnostic accuracy. Kaplan–Meier curves were employed for analyzing the association between CAB39L expression and OS, DSS, and PFS of KIRC patients. Survival advantages were calculated by log-rank (Mantel–Cox) test. The independent prognostic significance of clinical parameters for OS such as CAB39L in KIRC patients was estimated by univariate and multivariate Cox analysis. On this basis, a nomogram was established for predicting OS of KIRC patients at 1, 3, and 5 years. The efficiency of the nomogram for OS was validated using a calibration plot. The nomogram and calibration plot were conducted by the R package “rms”. Continuous data were exhibited in the style of “mean ± SD”. *p* < 0.05 represents statistical significance.

## 3. Results

### 3.1. Pan-Cancer Analysis of CAB39L Expression

The mRNA levels of CAB39L were determined in 33 types of cancer from independent databases. Firstly, the mRNA expression of CAB39L in TCGA database was investigated. CAB39L expression was markedly lower in tumor samples than normal kidney tissues, especially in kidney renal papillary cell carcinoma (KIRP), KIRC, and kidney chromophobe (KICH) (Figure 1a,b). The same expression trend was also observed in the TIMER database (Figure 1c). The above findings indicated that CAB39L may have a suppressive tumor-growth effect in most cancers.

### 3.2. Transcriptional Level of CAB39L in KIRC and Its Relationship with Clinical Features of KIRC Patients

Although overwhelming evidence suggests that CAB39L is a novel tumor biomarker, there is little research on the transcriptional analysis of CAB39L in KIRC. Hence, different databases were used to explore the expression of CAB39L and its association with clinicopathological paraments of KIRC patients. The mRNA levels of CAB39L were markedly downregulated in KIRC samples compared with normal kidney samples in TCGA database (Figure 2a). Then, the mRNA expression of CAB39L was analyzed in the M stage, N stage, T stage, pathologic stage, and histologic grade, as well as its association with clinicopathological parameters of KIRC patients. It was discovered that transcriptional levels of CAB39L were much lower in tissues of KIRC patients with advanced and metastatic cancer stages (Figure 2b–f). Moreover, CAB39L expression had a significant association with histological grade, pathologic stage, M stage, and T stage (Table 1). Logistic regression analysis confirmed that CAB39L expression was negatively corelated with various clinicopathological parameters of poor prognosis, including T stage, M stage, histologic grade, and pathologic stage (Table 2). CAB39L expression in KIRC was further analyzed by GEPIA and UALCAN databases to make the findings more reliable. Consistent with previous analysis results in TCGA database, lower mRNA expression of CAB39L similarly occurred in patients with advanced KIRC (Figure 3a–f). The above findings indicated that lower CAB39L expression may predict worse prognosis.

### 3.3. Diagnostic Value of CAB39L Expression in KIRC

The diagnostic value of CAB39L mRNA expression for various clinical characteristics of KIRC patients was evaluated using ROC curves. The AUC value was 0.9 for T1 stage, 0.889 for T2 stage, 0.959 for T3 stage, 0.949 for T4 stage, 0.921 for N0 stage, 0.957 for N1 stage, 0.916 for M0 stage, 0.955 for M1 stage, 0.894 for I and II stage, 0.958 for III and IV stage, 0.895 for G1 and G2 stage, and 0.947 for G3 and G4 stage (Figure 4a–c). The upper results showed that the transcriptional level of CAB39L was relatively specific and sensitive for KIRC diagnosis.

### 3.4. Prognostic Value of CAB39L Expression in KIRC

Kaplan–Meier curves were used to verify the prediction of CAB39L mRNA level for clinical outcomes. As shown in Figure 4d–f, OS (HR = 0.41, *p* < 0.001), PFS (HR = 0.44, *p* < 0.001), and DSS (HR = 0.30, *p* < 0.001) for KIRC patients with relatively low CAB39L expression were all significantly worse than those with high CAB39L expression. The effect of CAB39L expression on OS in KIRC patients with various stages and clinical parameters was also evaluated (Figure 5a). Univariate and multivariate Cox regression analyses were further applied to explore the independent prognostic value of CAB39L (Table 3). The results showed that the lower mRNA expression of CAB39L was independently related to a significantly shorter OS (HR = 0.6, 95% CI: 0.375–0.962, *p* = 0.034). On this basis, the OS of KIRC patients at 1, 3, and 5 years was predicted using the nomogram model (Figure 5b), and the efficiency of the nomogram for OS was validated by calibration curves (Figure 5c). The nomogram included age, M stage, and CAB39L, with a C-index of 0.725. The above results demonstrated that the transcriptional level of CAB39L had a strong prognostic value.

### 3.5. Construction of the PPI Network and Enrichment Analysis of CAB39L-Related Genes and DEGs

The PPI network of CAB39L and its associated genes with similar functions was constructed to elucidate the potential interaction using the STRING tool (Figure 6a). CAB39L-related genes included STK11, STRADA, SREADB, STK24, CAB39, PRKAA1, PRKAA2, PRKAB2, and PRKAG2. Combined with GO and KEGG analysis of CAB39L-related genes (Figure 6b), it was found that CAB39L and its related genes were primarily associated with protein kinase activity, protein kinase complex, protein serine/threonine kinase activity, and the AMPK signaling pathway. Furthermore, to investigate the potential mechanism of CAB39L in the progression of KIRC, KIRC samples were divided into low and high CAB39L groups to identify CAB39L-related DEGs (Figure 6c). The top 15 upregulated and downregulated DEGs (Appendix A) were chosen to perform GO and KEGG analysis (Figure 6d). The results indicated that CAB39L-related DEGs were enriched in oxidative phosphorylation, ATPase activity, and proton-exporting ATPase complex.

### 3.6. CAB39L Co-Expression Network in KIRC

Co-expressed genes with CAB39L were searched from LinkedOmics and exhibited using volcano maps (Figure 7a). The top 50 genes negatively and positively corelated with CAB39L were displayed using heatmaps (Figure 7b,c). GO analysis indicated that CAB39L co-expressed genes were enriched in the tricarboxylic acid metabolic process, NADH dehydrogenase complex assembly, and mitochondrial respiratory chain complex assembly (Figure 7d). KEGG analysis indicated the enrichment of oxidative phosphorylation, propanoate metabolism, and the TCA cycle (Figure 7e). Combined with the previous analysis of CAB39L-related genes and DEGs, it was speculated that the function of CAB39L could be correlated with substance and energy metabolism in KIRC.

### 3.7. Methylation Level of CAB39L Gene Promoter Region in KIRC

DNA methylation modification is one of the most important components of epigenetics and can regulate gene expression. To investigate the cause of low CAB39L expression in KIRC, the methylation level of CAB39L was analyzed through the UALCAN database. Compared with normal tissues, CAB39L was hypermethylated in KIRC tissues (Figure 8a). The level of CAB39L gene methylation increased with the progression of tumor grade and stage (Figure 8b,c). In addition, the methylation level of different CpG sites located at the CAB39L gene promoter region was analyzed using the MethSurv web tool (Figure 8d). Consistent with our previous results, almost all cpG sites had a high methylation level. The above results demonstrated that the downregulation of CAB39L in KIRC was possibly due to promoter hypermethylation, which was positively correlated with tumor grade and stage.

### 3.8. CAB39L Protein Expression in Paired KIRC Samples

WB was chosen to detect CAB39L protein expression in KIRC tissues for validating the results of TCGA, GEPIA, and UALCAN databases (Figure 9a). Compared with paracancerous tissues, the protein level of CAB39L was markedly downregulated in KIRC tissues. The protein level of CAB39L in clinical KIRC tissues and the HPA database was also detected by IHC, and the results of IHC were in accordance with those of Western blot (Figure 9b,c).

### 3.9. Overexpression of CAB39L Inhibited Tumorigenicity and Metastasis of KIRC Cells In Vitro

The expression of CAB39L was overexpressed using plasmid to explore the impact of CAB39L on the proliferation of KIRC cells. The overexpression of CAB39L significantly impaired the proliferative capability of A498 and 786-O cells, which was confirmed by CCK-8 and colony formation assays (Figure 10a–c). Furthermore, transwell migration and invasion assays revealed that CAB39L restrained the metastatic capacity of A498 and 786-O cells (Figure 10d). In conclusion, the above results suggested that CAB39L impaired the tumorigenesis of KIRC.

## 4. Discussion

KIRC is the most common pathological subtype of RCC [1]. Most patients with KIRC remain asymptomatic at an early stage, and more than 60% of patients with KIRC are incidentally detected by imaging examinations for other reasons [1,2]. However, the survival prognosis of patients with advanced KIRC is far from satisfactory, despite the landmark improvement in diagnosis and treatment [2,4]. Thus, there is an urgent need to discover a sensitive and effective biomarker for KIRC.

CAB39L is a scaffold protein that can interact with STRAD pseudokinase to activate LKB1 tumor suppressive kinase activity [10,24]. Recently, several studies have reported that downregulated CAB39L expression is significantly correlated to poor prognosis and genesis and development of malignant cancers [11,12,25]. In breast cancer and gastric carcinoma, silencing CAB39L facilitated G1/S phase transition and reduced cell apoptosis and thus promoted tumor progression, while overexpression of CAB39L exerted opposite effects [11,12]. However, the exact function of CAB39L in KIRC is still unclear. Hence, the expression profile, diagnostic and prognostic significance, interactive network, and potential mechanism of CAB39L in KIRC were systematically analyzed in this study.

The transcriptional level of CAB39L and KIRC across independent databases (TCGA, TIMER, GEPIA, and UALCAN) was investigated, indicating that CAB39L expression was lower in KIRC than in paracancerous samples. This finding was also confirmed by a decrease in the protein expression level of CAB39L detected by WB and IHC. Interestingly, the results also revealed that the mRNA level of CAB39L was downregulated in most cancers, including breast cancer, gastric cancer, and bladder cancer. Subsequently, the cause of the downregulation of CAB39L in KIRC was explored to discover that the methylation level of the CAB39L gene promoter region was significantly elevated. Taken together, CAB39L may play a potential suppressive role in tumor development.

In addition, the correlation between CAB39L expression and the clinical features of KIRC patients was further examined in this experiment. The results revealed that CAB39L expression was closely associated with tumor stage and histologic grade. Logistic regression analysis further demonstrated that CAB39L expression was negatively correlated with pathologic stage, histologic grade, and M and T stage. The Kaplan–Meier curves exhibited that low CAB39L expression suggested unfavorable OS, PFS, and DSS in KIRC patients. Moreover, a low CAB39L mRNA level was an independent undesirable prognostic parameter for OS in KIRC, according to univariate and multivariate regression analyses. The ROC curve showed that CAB39L expression had a strong diagnostic value, regardless of early or late KIRC. Owing to the significant prognostic and diagnostic value of the mRNA level of CAB39L, a nomogram based on CAB39L expression and some clinical characteristics was constructed to assist in predicting the mortality risk and optimizing the clinical decision.

To figure out the potential mechanism of CAB39L in KIRC, a PPI network analysis of co-regulatory proteins of CAB39L was first performed, finding that CAB39L-related genes were mainly associated with the AMPK signaling pathway and protein kinase activity. AMPK is a crucial metabolic checkpoint for maintaining cellular energy balance and regulating tumor growth according to energy signals [26,27]. Some studies reported that activated AMPK could promote oxidative phosphorylation in mitochondria and restrain aerobic glycolysis (Warburg effect) to repress tumorigenesis [26,28,29]. Interestingly, it was further discovered that CAB39L-related DEGs were significantly related to oxidative phosphorylation and ATP production. Moreover, the enrichment analysis of CAB39L co-expression genes indicated that CAB39L participated in regulating oxidative phosphorylation, mitochondrial respiratory chain, and the tricarboxylic acid cycle. Metabolic dysregulation exists in many cancers, which is mainly characterized by the Warburg effect [30,31,32,33]. The majority of cancer cells still choose a high glycolytic metabolism to meet the requirement of uncontrolled cell proliferation, even in the presence of adequate oxygen [33]. Taken together, it can be assumed that CAB39L may shift aerobic glycolysis to oxidative phosphorylation through the AMPK signal pathway to limit tumor growth.

Moreover, the effect of CAB39L expression on the malignant phenotype of KIRC cells was explored in vitro. The results indicated that the overexpression of CAB39L significantly impaired the proliferative and metastatic capacity of 786-O and A498 cells. These above results further suggested that CAB39L was positively involved in repressing the genesis and development of KIRC cells. Whether CAB39L regulates tumor growth by influencing the cell cycle in KIRC needs further experimental verification. However, there were some limitations in this study. First, the function of CAB39L was not confirmed in vivo. Second, the prognostic and diagnostic value of CAB39L was only investigated and validated in cancer databases, requiring more clinical research support. Finally, although the potential downstream mechanism of CAB39L was analyzed through bioinformatics, its underlying molecular mechanism was not verified both in vitro and in vivo. Therefore, it is necessary to investigate the possible mechanism of CAB39L in KIRC in subsequent studies.

## 5. Conclusions

CAB39L is a critical indicator for diagnosing and predicting the prognosis of KIRC patients. Moreover, CAB39L is downregulated in both the protein and mRNA level of KIRC and plays an indispensable role in restraining tumor progression. This study may provide new insights into improving the diagnosis, prognosis, and clinical decisions of KIRC patients.

## Figures and Tables

**Figure 1 medicina-59-00716-f001:**
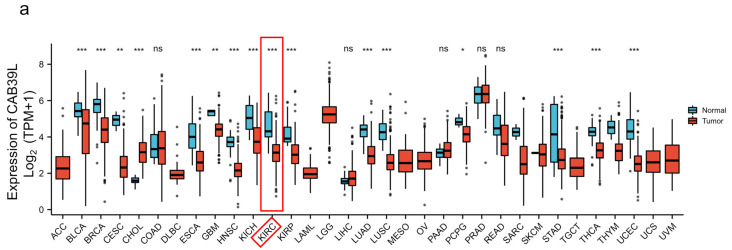
The mRNA level of CAB39L in different types of cancer. (**a**) mRNA expression of CAB39L in 33 types of cancer (TCGA). (**b**) mRNA expression of CAB39L in tumor and paired paracancerous samples (TCGA). (**c**) mRNA expression of CAB39L in 33 types of cancer (TIMER). ACC: adrenocortical carcinoma; BLCA: bladder urothelial carcinoma; BRCA: breast invasive carcinoma; CESC: cervical squamous cell carcinoma and endocervical adenocarcinoma; CHOL: cholangiocarcinoma; COAD: colon adenocarcinoma; DLBC: lymphoid neoplasm diffuse large B-cell lymphoma; ESCA: esophageal carcinoma; GBM: glioblastoma multiforme; HNSC: head and neck squamous cell carcinoma; KICH: kidney chromophobe; KIRC: kidney renal clear cell carcinoma; KIRP: kidney renal papillary cell carcinoma; LAML: acute myeloid leukemia; LGG: brain lower grade glioma; LIHC: liver hepatocellular carcinoma; LUAD: lung adenocarcinoma; LUSC: lung squamous cell carcinoma; MESO: mesothelioma; OV: ovarian serous cystadenocarcinoma; PAAD: pancreatic adenocarcinoma; PCPG: pheochromocytoma and paraganglioma; PRAD: prostate adenocarcinoma; READ: rectum adenocarcinoma; SARC: sarcoma; SKCM: skin cutaneous melanoma; STAD: stomach adenocarcinoma; TGCT: testicular germ cell tumors; THCA: thyroid carcinoma; THYM: thymoma; UCEC: uterine corpus endometrial carcinoma; UCS: uterine carcinosarcoma; UVM: uveal melanoma. * *p* < 0.05, ** *p* < 0.01, *** *p* < 0.001, ns: no significance.

**Figure 2 medicina-59-00716-f002:**
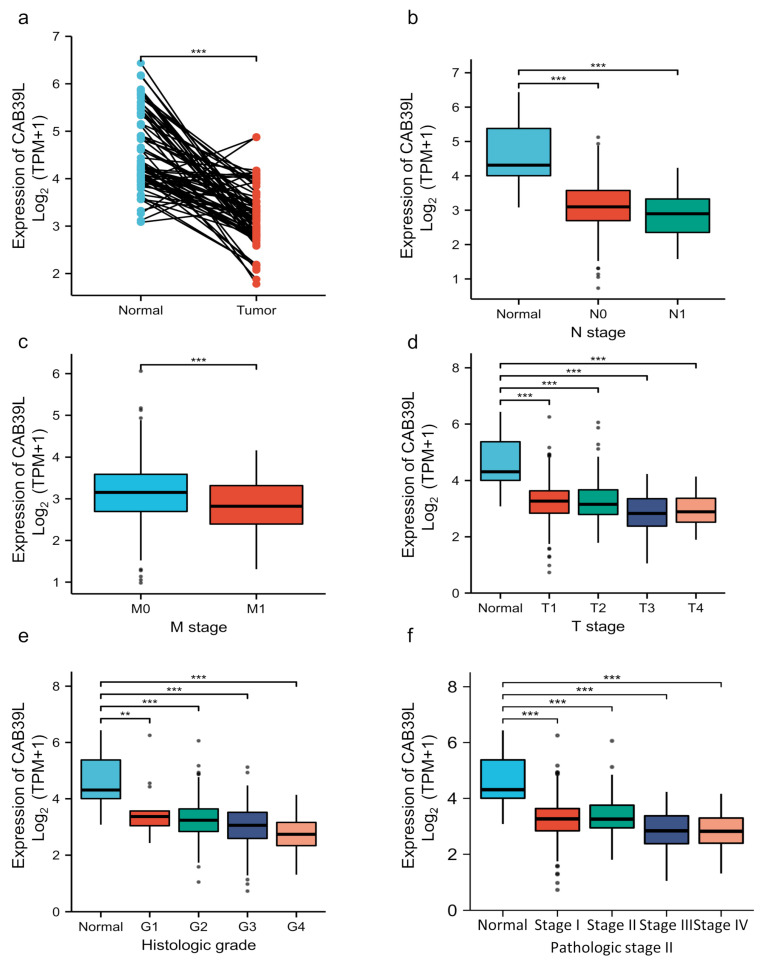
Analysis of the transcriptional level of CAB39L and its association with clinical parameters of KIRC patients (TCGA). (**a**) mRNA expression of CAB39L in 72 KIRC samples and adjacent non-tumor samples. (**b**) Relationship between the mRNA expression of CAB39L and N stage. (**c**) Relationship between the mRNA expression of CAB39L and M stage. (**d**) Relationship between the mRNA expression of CAB39L and T stage. (**e**) Relationship between the mRNA expression of CAB39L and histologic grade. (**f**) Relationship between the mRNA expression of CAB39L and pathologic stage. T: tumor size; M: distant metastasis; N: lymphatic metastasis; ** *p* < 0.01, *** *p* < 0.001.

**Figure 3 medicina-59-00716-f003:**
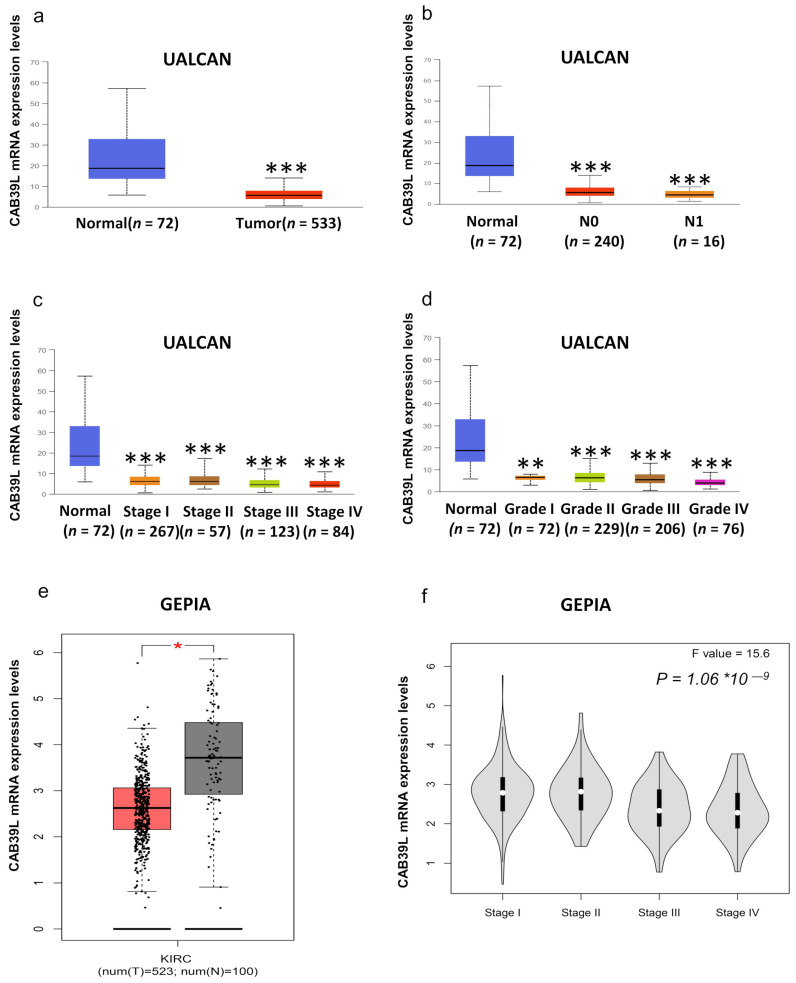
Analysis of the transcriptional level of CAB39L and its association with clinical parameters of KIRC patients (UALCAN and GEPIA). (**a**) mRNA expression of CAB39L in 533 KIRC samples and 72 normal tissue samples (UALCAN). (**b**–**d**) Relationship between the transcriptional level of CAB39L and N stage, pathologic stage, and histologic grade (UALCAN). (**e**) The transcriptional level of CAB39L in 523 KIRC samples and 100 normal tissue samples (GEPIA). (**f**) Relationship between the transcriptional level of CAB39L and pathologic stage (GEPIA). * *p* < 0.05, ** *p* < 0.01, *** *p* < 0.001.

**Figure 4 medicina-59-00716-f004:**
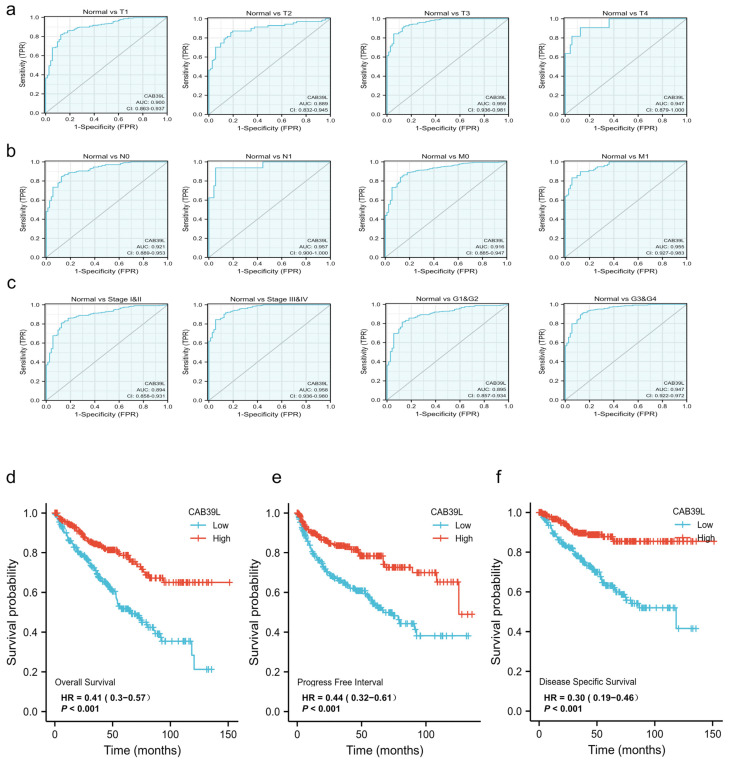
Diagnostic and prognostic capacity of CAB39L in KIRC. (**a**) Diagnostic capacity of CAB39L expression in KIRC patients at T stage by ROC curves. (**b**) Diagnostic capacity of CAB39L expression in KIRC patients at M and N stages by ROC curves. (**c**) Diagnostic capacity of CAB39L expression in KIRC patients at different histologic grade and pathologic stage by ROC curves. (**d**) Association between CAB39L expression and OS (TCGA). (**e**) Association between CAB39L expression and PFS (TCGA). (**f**) Association between CAB39L expression and DSS (TCGA). Progression-free survival (PFS), disease-specific survival (DSS), overall survival (OS).

**Figure 5 medicina-59-00716-f005:**
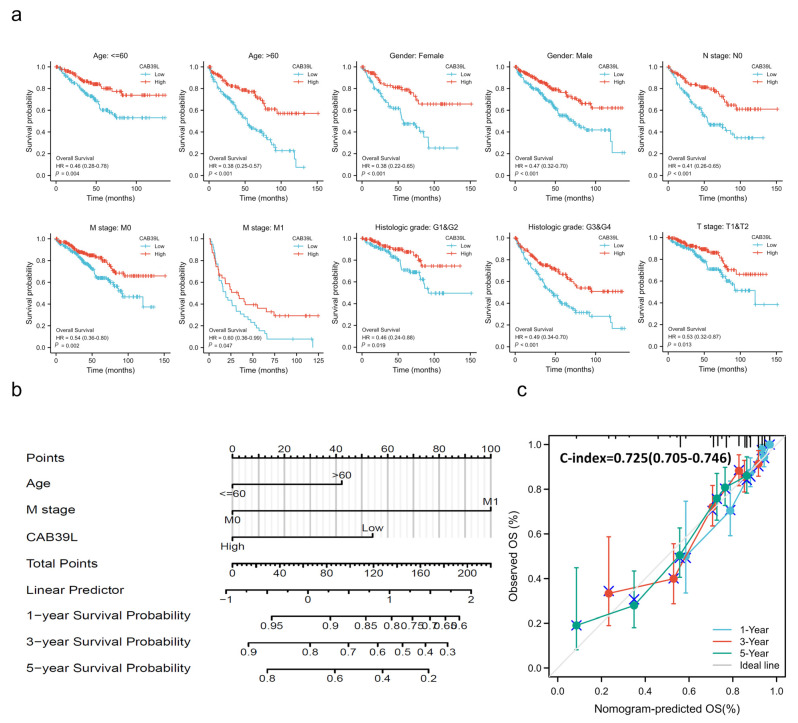
Prognostic value of CAB39L in subgroups of KIRC patients and construction and validation of the nomogram based on CAB39L mRNA expression. (**a**) The relationship between CAB39L expression and OS in different subgroups, including age ≤ 60, age > 60, female, male, N0 stage, M0 stage, M1 stage, T1 and T2 stage, G1 and G2 stage, and G3 and G4 stage. (**b**) The OS of KIRC patients at 1, 3, and 5 years was predicted by the nomogram model. (**c**) The efficiency of the nomogram for OS was validated by calibration curves.

**Figure 6 medicina-59-00716-f006:**
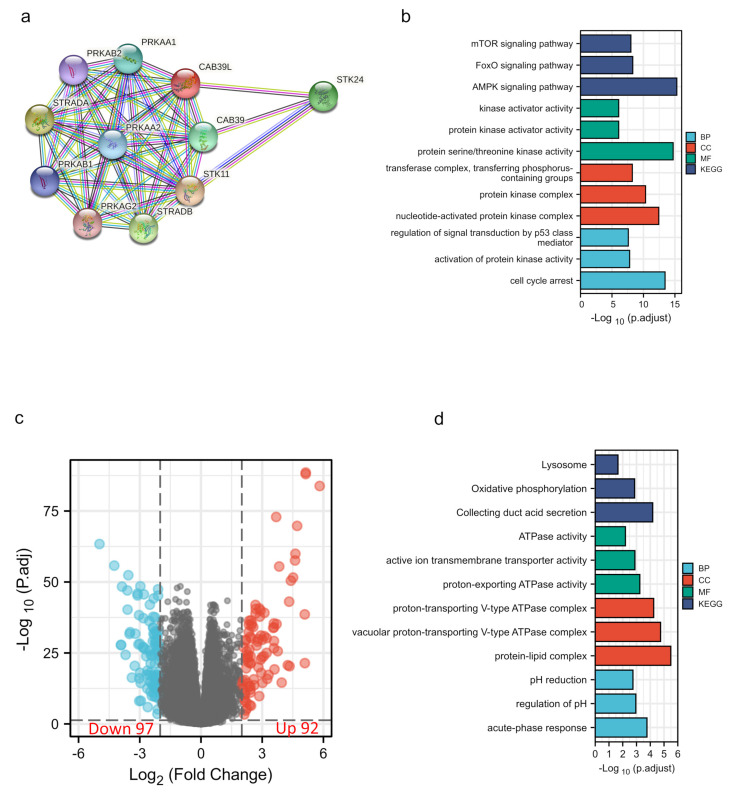
PPI construction and enrichment analysis of CAB39L. (**a**) A web of CAB39L and its potential co-expressed genes constructed using STRING. (**b**) GO and KEGG enrichment analysis performed on CAB39L-related genes. (**c**) Volcano plot of DEGs between the expression of high and low CAB39L in KIRC (|log2fold change| > 2 and adjusted *p* < 0.05). (**d**) GO and KEGG enrichment analysis of CAB39L-related DEGs.

**Figure 7 medicina-59-00716-f007:**
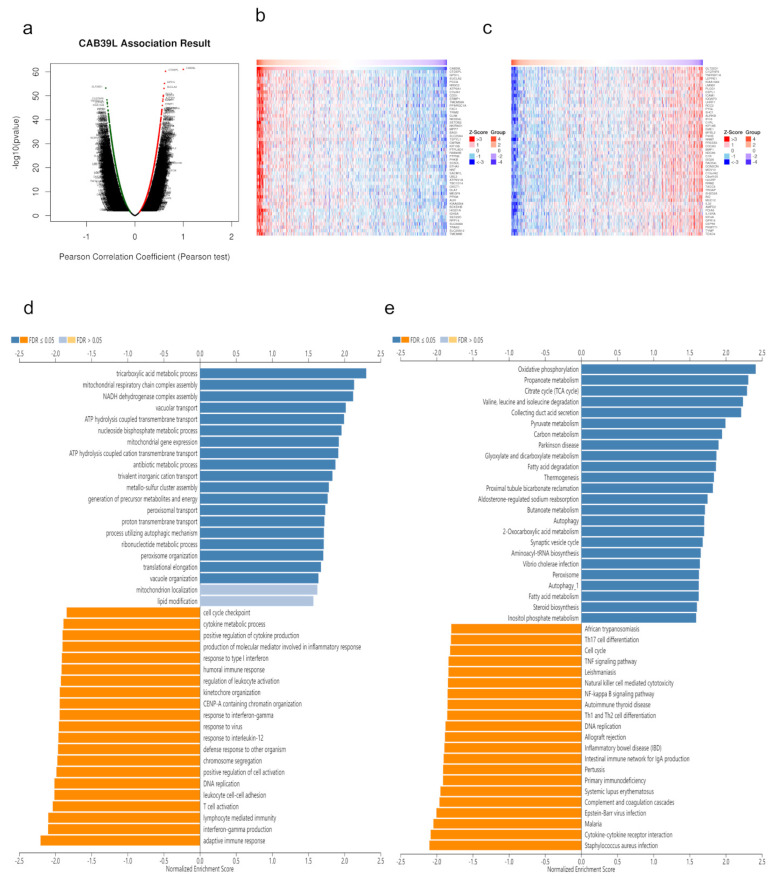
CAB39L co-expressed genes and functional enrichment analysis (LinkedOmics). (**a**) Volcano map of CAB39L co-expressed genes in KIRC. (**b**) Fifty genes positively associated with CAB39L. (**c**) Fifty genes negatively associated with CAB39L. (**d**) GO enrichment analysis of CAB39L co-expressed genes. (**e**) KEGG enrichment analysis of CAB39L co-expressed genes.

**Figure 8 medicina-59-00716-f008:**
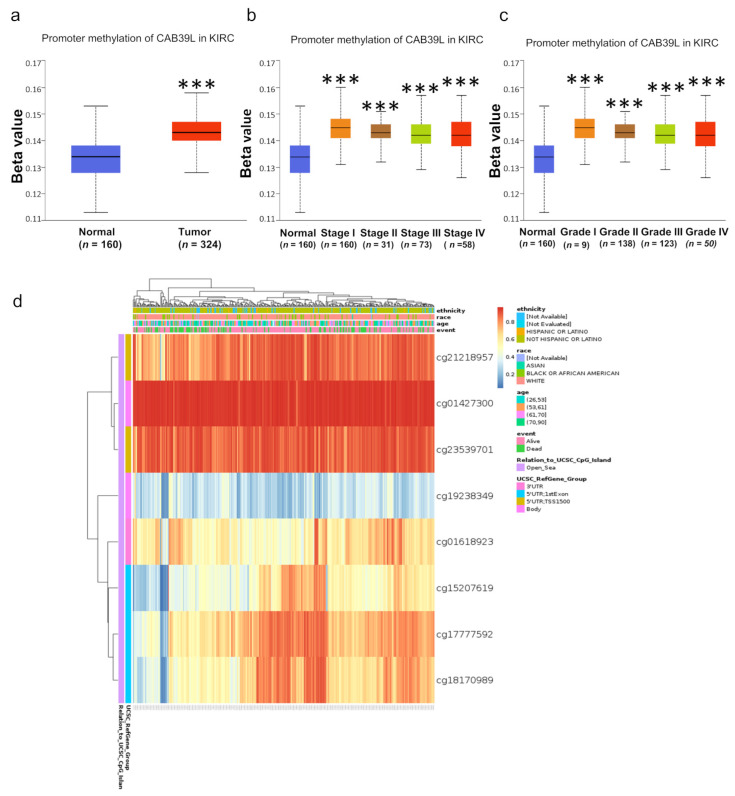
Analysis of methylation level in the CAB39L gene promoter region. (**a**) Analysis of methylation level of the CAB39L gene promoter region for KIRC compared with normal tissues (UALCAN). (**b**,**c**) Analysis of the relationship between methylation level of the CAB39L promoter region and pathologic stages (UALCAN). (**c**) Analysis of the relationship between methylation level of the CAB39L promoter region and histologic grades (UALCAN). (**d**) Analysis of methylation level of the CAB39L promoter region at CpG sites (MethSurv). *** *p* < 0.001.

**Figure 9 medicina-59-00716-f009:**
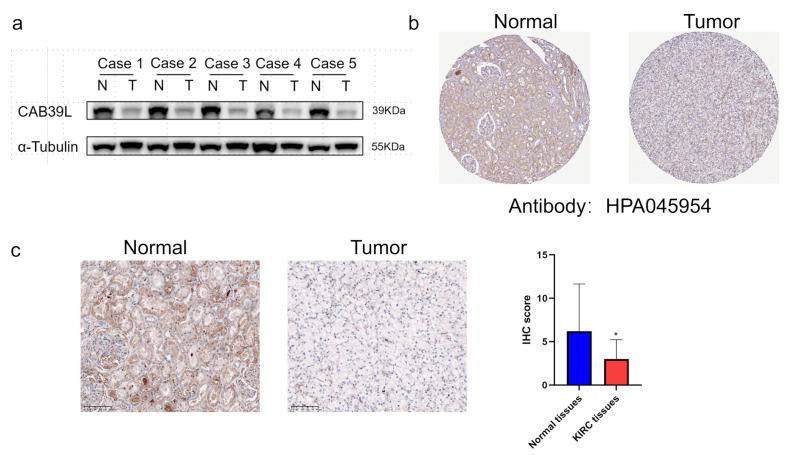
The protein level of CAB39L in paired KIRC samples. (**a**) Western blot detection of CAB39L expression in five paired KIRC samples. (**b**) IHC staining outcomes of CAB39L in KIRC and adjacent non-tumor tissues (HPA). (**c**) IHC staining outcomes of CAB39L in KIRC and adjacent non-tumor tissues (clinical paired KIRC tissues). * *p* < 0.05.

**Figure 10 medicina-59-00716-f010:**
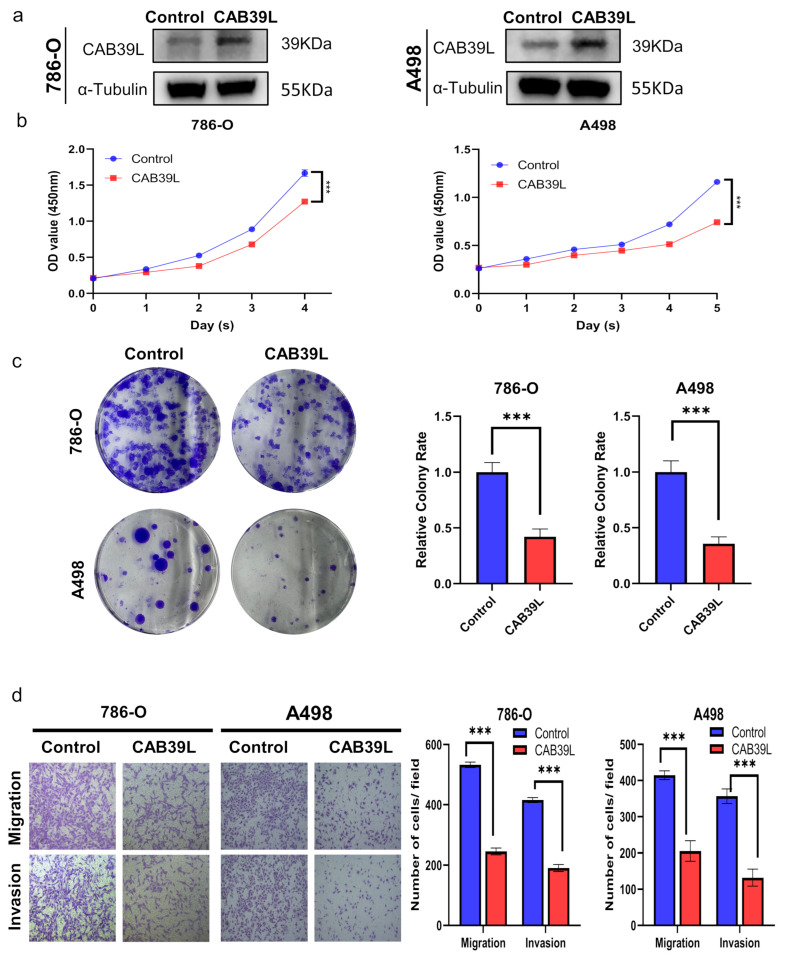
Overexpression of CAB39L inhibited the tumorigenicity and metastasis of KIRC cells in vitro. (**a**) Western blot detection of CAB39L expression after transfection of CAB39L overexpression plasmid in 786-O and A498 cells. (**b**) Impact of CAB39L overexpression on cell proliferation detected by CCK-8 assay. (**c**) Impact of CAB39L overexpression on cell proliferation detected by colony formation assay. (**d**) Impact of CAB39L overexpression on migration and invasion capacity of 786-O and A498 cells detected by transwell assays. *** *p* < 0.001.

**Table 1 medicina-59-00716-t001:** Association of CAB39L mRNA expression with clinical characteristics in KIRC patients.

Characteristic	Low Expression of CAB39L *(n* = 269)	High Expression of CAB39L *(n* = 270)	*p*
**Age, *n* (%)**			0.897
<=60	133 (24.7%)	136 (25.2%)	
>60	136 (25.2%)	134 (24.9%)	
**Gender, *n* (%)**			0.334
Female	87 (16.1%)	99 (18.4%)	
Male	182 (33.8%)	171 (31.7%)	
**T stage, *n* (%)**			**<0.001**
T1	111 (20.6%)	167 (31%)	
T2	34 (6.3%)	37 (6.9%)	
T3	117 (21.7%)	62 (11.5%)	
T4	7 (1.3%)	4 (0.7%)	
**N stage, *n* (%)**			0.571
N0	125 (48.6%)	116 (45.1%)	
N1	10 (3.9%)	6 (2.3%)	
**M stage, *n* (%)**			**0.005**
M0	203 (40.1%)	225 (44.5%)	
M1	51 (10.1%)	27 (5.3%)	
**Histologic grade, *n* (%)**			**<0.001**
G1	5 (0.9%)	9 (1.7%)	
G2	95 (17.9%)	140 (26.4%)	
G3	112 (21.1%)	95 (17.9%)	
G4	56 (10.5%)	19 (3.6%)	
**Pathologic stage, *n* (%)**			**<0.001**
Stage I	108 (20.1%)	164 (30.6%)	
Stage II	25 (4.7%)	34 (6.3%)	
Stage III	79 (14.7%)	44 (8.2%)	
Stage IV	55 (10.3%)	27 (5%)	

**Table 2 medicina-59-00716-t002:** Logistic analysis of the association between CAB39L mRNA expression and clinical characteristics of KIRC patients.

Characteristics	Total (N)	Odds Ratio (OR)	*p*-Value
Age (>60 vs. ≤60)	539	0.964 (0.687–1.351)	0.829
Gender (Male vs. Female)	539	0.826 (0.578–1.178)	0.291
T stage (T3 and T4 vs. T1 and T2)	539	0.378 (0.261–0.544)	**<0.001**
N stage (N1 vs. N0)	257	0.647 (0.214–1.797)	0.413
M stage (M1 vs. M0)	506	0.478 (0.285–0.784)	**0.004**
Histologic grade (G3 and G4 vs. G1 and G2)	531	0.455 (0.321–0.644)	**<0.001**
Pathologic stage (Stage III and Stage IV vs. Stage I and Stage II)	536	0.356 (0.247–0.510)	**<0.001**

**Table 3 medicina-59-00716-t003:** Cox regression analysis of variables for OS in KIRC patients.

Characteristics	Total (N)	Univariate Analysis		Multivariate Analysis	
		Hazard Ratio (95% CI)	*p*-Value	Hazard Ratio (95% CI)	*p*-Value
**Age**					
≤60	269	Reference			
>60	270	1.765 (1.298–2.398)	**<0.001**	1.617 (1.054–2.482)	**0.028**
**Gender**					
Female	186	Reference			
Male	353	0.930 (0.682–1.268)	0.648		
T stage					
T1 and T2	349	Reference			
T3 and T4	190	3.228 (2.382–4.374)	**<0.001**	1.497 (0.658–3.406)	0.336
N stage					
N0	241	Reference			
N1	16	3.453 (1.832–6.508)	**<0.001**	1.602 (0.797–3.222)	0.186
**M stage**					
M0	428	Reference			
M1	78	4.389 (3.212–5.999)	**<0.001**	2.761 (1.632–4.671)	**<0.001**
**Histologic grade**					
G1 and G2	249	Reference			
G3 and G4	282	2.702 (1.918–3.807)	**<0.001**	1.556 (0.937–2.583)	0.088
**Pathologic stage**					
Stage I and Stage II	331	Reference			
Stage III and Stage IV	205	3.946 (2.872–5.423)	**<0.001**	1.230 (0.487–3.104)	0.662
**CAB39L**					
Low	270	Reference			
High	269	0.415 (0.300–0.574)	**<0.001**	0.600 (0.375–0.962)	**0.034**

## Data Availability

The data that support the findings of this study are available from the corresponding author upon request.

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
