# Peer review of "Bioinformatics Prediction and Experimental Verification Identify CAB39L as a Diagnostic and Prognostic Biomarker of Kidney Renal Clear Cell Carcinoma"

_medicina, 2023, doi:10.3390/medicina59040716_

Round 1

Reviewer 1 Report

The article evaluates the diagnostic and prognostic value of CAB39L, which is a novel tool, not previously reported. I consider that the level of originality and scientific interest is very high. Yet, the experimental research needs clinical validation.

Considering the possibility of protein CAB39L evaluation by IHC and the nomogram which was proposed by the authors, it could be an useful tool for clinical decisions. The findings deserve further validation as it seems to be a promising and not so complicated instrument.

Reviewer 2 Report

The overall impression of the manuscript is that it is overcrowded. Authors include a large number of figures (10! and each one with several inside), many of which could easily fit within the Appendix, making the flow of the reading easier and the presentation clearer.

Some specific questions that were not clear to me are:

In Materials and Methods, line 80, was the expression data transformed?, z-scored? etc? Also there, why was the median chosen to separate both groups. Is there a plot to show a clear separation between both groups? or there is a groups of samples that are in very close to the mean? PErhaps it would have been better to choose other cutoffs such as top and bottom quartiles, or similar, even if it will mean having fewers samples, it could benefit the group assignment and separation. 

Figures must be improved. Several of the axis labels are not ok. For example in Figure 1 only in the c plot it is correct, remove The in the label for a and b. 

Figure 2, remove "The" in labels.

Figure 3: add levels after expression in the y labels.

Figure 4 a, b, ...f, are too large but the size of the numbers and labels in a b, and c figures are too small.

Figure 5 similar to 4

Figure 7 similat to 4, not possible to red the variables, or the pathways. Are all these figures really necessary to be in the main text? Heatmaps should probably go to the appendix in much larger size.

Figure 8, again problematic, extremely large a, b, c,and d yet the legend and the axes are too small be be read. 

Figure 9 huge font, not proportional to the rest of the figure or the boxplot font sizes or the text. 

Figure 10 again problems with the figures, the one in d right has been extended, clearly to see it form the font, please put it to the right scale. And make the a, b, c, and d to be normal sizes. 

Some spelling mistakes and grammar should be corrected, some examples below:  

In the Abstract line 26, low diagnostic value despite early or late KIRC, rephrase it, do you mean in both? 

Remove the How to Use this template section. 

Line 59: More and more, one extra capital M

In the Introduction lines 63 to 69 are conclusions and not introduction. 

Line 106: run, not runed

Line 123: 200 preheated cells, not Preheated 200 cells

Line 125 500 pretreated cells were seeded and not preheated 500 cells were seeded

Line 125: rephrase as for colony

Line 137: complete lysing

Line 152: in this experiment all signed (remove were)

Line 155: in detail instead of specifically

Line 171: change means for another word

Line 197: levels of

Line 199: histological

Lines 267 and 287: showed that instead of exhibited that

Be consistent, sometimes you add a space in the pvalue after the < sign, other you do not.

Line 332: enriched and not concentrated

Reviewer 3 Report

The manuscript by Yunfei Wu and co-workers explored bioinformatics prediction and experimental verification identify CAB39L as a diagnostic and prognostic biomarker of kidney renal clear cell carcinoma (KIRC). They evaluated CAB39L possesses prognostic and diagnostic capacity in KIRC. My overall evaluation of the manuscript is positive. There are a number of minor revisions, formal and scientific aspects that should be addressed.

1.                  Take more detail in the introduction and discussion about the effect of CAB39L silencing on the cell cycle in KIRC.

2.                  In the section of cell culture and transfection, it is necessary to fully explain the working method. It is not clear how long incubation has been done after the cultivation of cells? And what was the confluence at the time of transfection. How is the transfection done? What concentration of transfection material and vector was used?

3.                  In the western blot part, it is not clear whether the antibodies were prepared in the blocking solution or in TBST, TBS or PBS? What characteristics did the secondary antibody have, for example, with what labels were it conjugated? For example, Alexa Fluor®, HRP, AP, Biotin, IRDye®. What combination was used to stain the protein bands?

4.                  Why didn't you use GAPDH as an internal western control? Explain the advantage of tubulin in this cancer.

5.                  The results are not clearly described. It is better according to the extensive statistical analysis that has been done. Statistical information should be converted into biological language to enable the use of these data. Therefore, it is necessary to rewrite the results again.

6.                  In Figure 10, parts C and D, there seems to be a discrepancy between the control and treatment groups. That is, the control is not related to the same treatment that is shown. It must be replaced with correct images.

7.                  The description of Figure 1 and 2 are insufficient and it is necessary to briefly explain the things described in the text below Figure 1 and 2.
